# Polyamine Homeostasis in Snyder-Robinson Syndrome

**DOI:** 10.3390/medsci6040112

**Published:** 2018-12-07

**Authors:** Tracy Murray-Stewart, Matthew Dunworth, Jackson R. Foley, Charles E. Schwartz, Robert A. Casero

**Affiliations:** 1Sidney Kimmel Comprehensive Cancer Center, Johns Hopkins University, Baltimore, MD 21287, USA; tmurray2@jhmi.edu (T.M.-S.); matthewdunworth@jhmi.edu (M.D.); jfoley13@jhmi.edu (J.R.F.); 2The Greenwood Genetic Center, Greenwood, SC 29646, USA; ceschwartz@ggc.org

**Keywords:** Snyder-Robinson Syndrome, spermine synthase, X-linked intellectual disability, polyamine transport, spermidine, spermine, transglutaminase

## Abstract

Loss-of-function mutations of the spermine synthase gene (*SMS*) result in Snyder-Robinson Syndrome (SRS), a recessive X-linked syndrome characterized by intellectual disability, osteoporosis, hypotonia, speech abnormalities, kyphoscoliosis, and seizures. As SMS catalyzes the biosynthesis of the polyamine spermine from its precursor spermidine, SMS deficiency causes a lack of spermine with an accumulation of spermidine. As polyamines, spermine, and spermidine play essential cellular roles that require tight homeostatic control to ensure normal cell growth, differentiation, and survival. Using patient-derived lymphoblast cell lines, we sought to comprehensively investigate the effects of SMS deficiency on polyamine homeostatic mechanisms including polyamine biosynthetic and catabolic enzymes, derivatives of the natural polyamines, and polyamine transport activity. In addition to decreased spermine and increased spermidine in SRS cells, ornithine decarboxylase activity and its product putrescine were significantly decreased. Treatment of SRS cells with exogenous spermine revealed that polyamine transport was active, as the cells accumulated spermine, decreased their spermidine level, and established a spermidine-to-spermine ratio within the range of wildtype cells. SRS cells also demonstrated elevated levels of tissue transglutaminase, a change associated with certain neurodegenerative diseases. These studies form a basis for further investigations into the leading biochemical changes and properties of *SMS*-mutant cells that potentially represent therapeutic targets for the treatment of Snyder-Robinson Syndrome.

## 1. Introduction

First described in 1969 [1], Snyder-Robinson Syndrome (SRS) is an X-linked intellectual disability syndrome resulting from mutation of the spermine synthase (*SMS*) gene, located at chromosome Xp22.11 [2]. Active only as a homodimer [3], SMS catalyzes the production of spermine (SPM) from its precursor, spermidine (SPD), via the transfer of an aminopropyl group, which is derived from decarboxylated S-adenosylmethionine (dcAdoMet) through the action of S-adenosylmethionine decarboxylase (AdoMetDC; Figure 1). SRS males with the most severe phenotypes lack functional SMS protein, biochemically resulting in elevated levels of intracellular spermidine and near complete depletion of spermine. Spermidine and spermine, along with their precursor putrescine (PUT), constitute the mammalian polyamines, organic polycations that are absolutely essential for growth and proliferation. As their amine groups are protonated at physiological pH, polyamines interact with negatively charged intracellular moieties, including nucleic acids, chromatin, ion channels, certain proteins, and phospholipids [4,5,6,7]. Thus, alterations in intracellular polyamine concentrations can elicit potentially detrimental effects, and polyamine homeostasis must be tightly regulated through biosynthesis, catabolism, uptake, and excretion. Additionally, the primary amino groups of polyamines are natural substrates for transglutaminase-catalyzed reactions that result in protein cross-linking that has been associated with a number of pathologies [8,9]. As polyamines have essential roles in growth, differentiation, and development, the imbalance that occurs in SRS results in a combination of clinical manifestations including moderate-to-severe cognitive impairment, osteoporosis, asthenic build, low muscle mass, facial asymmetry, speech abnormalities, and seizures [10].

The current study investigates the biochemical effects of decreased SMS activity on the individual enzymes in polyamine metabolism as well as its effect on polyamine uptake from the extracellular environment and transglutaminase (TG) expression. SRS patient-derived lymphoblastoid cell lines are used that range in severity of *SMS* loss-of-function and spermine pool depletion, in comparison with those from healthy donors, to ascertain compensatory changes that might occur in an attempt to regulate polyamine homeostasis. Results of these studies provide useful background knowledge towards the goal of developing treatment strategies for these patients, of which there are currently none.

## 2. Materials and Methods 

### 2.1. Cell Lines and Culture Conditions

The lymphoblastoid cell lines were generated by transformation with Epstein–Barr virus as previously described [11,12,13]. The lines were derived from three SRS patients and two healthy male donors. Cells were grown in RPMI-1640 supplemented with 15% fetal bovine serum (Gemini Bio-Products, Sacramento, CA, USA), 2 mM glutamine, non-essential amino acids, sodium pyruvate, and penicillin/streptomycin in a humidified 5% CO_2_ atmosphere at 37 °C. Uptake experiments were conducted in the presence of 1 mM aminoguanidine (AG) to inhibit extracellular oxidation of spermine by bovine serum amine oxidase present in the culture medium. For these experiments, cells were incubated with either exogenous SPM (5 μM) or the polyamine analog bis(ethyl)norspermine (BENSpm) (10 μM) for 24 h prior to collection and preparation for HPLC analysis. BENSpm was synthesized as previously reported [14].

### 2.2. Assay of Polyamine Concentrations and Enzyme Activities

Cell lysates were acid extracted and labeled with dansyl chloride, followed by determination of intracellular polyamine concentrations via HPLC, as previously described [15]. Diaminoheptane, PUT, SPD, SPM, and acetylated derivatives of SPD and SPM used for HPLC standards were purchased from Sigma Chemical Co. (St. Louis, MO, USA). For HPLC analysis of culture medium, after 24 h of growth, each culture was pelleted and 2 mLs (1/5 of the total volume) of medium were removed, dried in a speed-vac, and resuspended in perchloric acid for acid extraction and labeling.

Enzyme activity assays were performed for spermidine/spermine *N*^1^-acetyltransferase (SSAT/*SAT1*), ornithine decarboxylase (ODC), and S-adenosylmethionine decarboxylase (AdoMetDC/*AMD1*) using radiolabeled substrates, as previously described [16,17,18]. Oxidation via spermine oxidase (SMOX) and *N*^1^-acetylpolyamine oxidase (PAOX) was measured using luminol-based detection of H_2_O_2_ in the presence of either SPM or *N*^1^-acetylated spermine (*N*^1^-AcSPM) as a substrate [19]. Enzyme activities and intracellular polyamine concentrations are presented relative to total protein in the lysate, which was determined by the method of Bradford [20], with interpolation on a bovine serum albumin standard curve. 

### 2.3. Protein Isolation and Western Blots

For Western blot analyses of proteins, cells were lysed in 4% SDS containing a protease inhibitor cocktail and homogenized using column-based centrifugation (Omega Bio-Tek, Norcross, GA, USA). The BioRad DC assay (Bio-Rad Laboratories, Hercules, CA, USA) was used for protein quantification. Equal amounts of reduced protein samples were separated on 4–12% Bis-Tris BOLT gels (Invitrogen, Carlsbad, CA, USA), transferred onto Immun-Blot PVDF (BioRad), and blocked in Odyssey blocking buffer (LI-COR, Lincoln, NE, USA). Primary antibodies were used targeting the following, ODC antizyme 1 (OAZ1) [21], spermidine synthase (SRM) (#19858-1-AP; Proteintech, Rosemont, IL, USA), histone deacetylase 10 (HDAC10) (#H3413; Sigma), and transglutaminase 2 (TGM2) (#ab421; Abcam, Cambridge, MA, USA), with pan histone H3 (#05-928; Upstate Cell Signaling Solutions, Lake Placid, NY, USA) as a normalization control. Secondary, species-specific, fluorophore-conjugated antibodies allowed visualization and quantification of bands via near-infrared imaging on an Odyssey detection system (LI-COR). Blot images were analyzed using Image Studio software (LI-COR, Lincoln, NE, USA).

### 2.4. RNA Isolation and Quantitation of Gene Expression

Total RNA was extracted from the lymphoblastoid cell lines using Trizol reagent (Invitrogen) and used for cDNA synthesis with qScript cDNA SuperMix (Quanta Biosciences, Gaithersburg, MD, USA). The mRNA expression levels of polyamine-metabolism-associated genes in the SRS versus wildtype (WT) lymphoblastoid lines were measured by SYBR-green-mediated (BioRad) quantitative real-time PCR on a BioRad iQ2 detection system. Custom primers specific for human *ODC1*, *OAZ1*, *AMD1*, *SRM*, *SMS*, *SAT1*, *HDAC10*, *SMOX*, *PAOX*, *TGM2*, and *GAPDH* were synthesized by Integrated DNA Technologies (Coralville, IA, USA). Primers were optimized on annealing temperature gradients with melt curve analyses and agarose gel electrophoresis. Triplicate determinations were obtained for each gene in each patient and normalized to *GAPDH* expression. The fold-change in expression was determined using the 2^-ΔΔCt^ algorithm.

### 2.5. Statistical Analyses

Statistically significant differences were determined by two-tailed Student’s *t*-tests with 95% confidence interval using GraphPad Prism software (La Jolla, CA, USA).

## 3. Results

### 3.1. Alterations in Intracellular Polyamine Distribution

It has been previously reported that spermine concentrations are reduced while spermidine concentrations are increased in SRS lymphoblast cell lines [11,12,13]. However, the overall effects on other enzymes in the polyamine pathway have not been thoroughly evaluated. Based on these previous studies, we chose three SRS cell lines with varying degrees of spermine synthase deficiency [10] (Table 1), which was confirmed by our HPLC analyses of intracellular polyamine concentrations (Figure 2). Along with SPM levels, PUT concentrations were also significantly decreased in the SRS lines relative to the WT lines, while the intracellular SPD pools significantly increased, as observed previously [11]. Consequently, the SPD/SPM ratio increased nearly 10-fold in the most affected lines (Table 1 and Figure 2). The total intracellular concentrations of polyamines did not significantly differ among the genotypes examined, regardless of the severity of spermine deficiency (Figure 2d), and none of the lysates contained detectable levels of acetylated SPD or SPM derivatives. 

### 3.2. Ornithine Decarboxylase Activity Is Decreased in Snyder-Robinson Syndrome 

Ornithine decarboxylase activity is the first rate-limiting enzyme in polyamine biosynthesis and catalyzes the production of PUT from ornithine (Figure 1). ODC activity was significantly lower in each of the SRS lymphoblast lines compared to WT controls, consistent with the reduction in PUT levels observed in these cells (Figure 3a). As the reductions in ODC activity did not correspond with reductions in ODC1 mRNA expression (Appendix A), we analyzed expression of ODC antizyme (OAZ1), a negative regulator of ODC protein that targets its degradation via the 26S proteasome. We consistently observed increased expression of OAZ1 protein only in SRS line 1, the least affected SRS line in terms of SMS activity and SPM depletion (Figure 3b). Although this increase might be responsible for the decreased ODC activity in these cells, it does not appear to contribute to that in SRS lines 2 or 3. Consequently, it is likely that product inhibition due to the increased levels of SPD plays a role in the reduced ODC activity. It is interesting that in spite of the obvious difference in SPD/SPM ratio between SRS1 (3.76) and the other 2 SRS lines (9.56 and 9.85), the decreases in ODC activity and PUT concentration among the three lines were quite similar, suggesting that the increased OAZ1 may serve to supplement the feedback regulation by SPD in SRS line 1. As with ODC, no apparent change in OAZ1 mRNA expression was observed to account for the change in protein (Appendix A).

### 3.3. Effects of Spermine Synthase Mutations on Spermidine Biosynthesis

Our current study and others have observed that a common result of *SMS* loss-of-function includes SPD accumulation (Figure 2b) [11,12,13,22,23,24]. SPD biosynthesis is similar to that of SPM: SRM catalyzes an aminopropyl group transfer from dcAdoMet to PUT, producing SPD (Figure 1). Both SRM and SMS aminopropyl transfer reactions are therefore limited by the availability of dcAdoMet and hence, the activity of AdoMetDC [25]. We examined the expression levels of these enzymes to determine the extent to which biosynthesis through these steps might be affected in SRS. We found that AdoMetDC activity and mRNA expression levels were similar among the five cell lines regardless of *SMS* status or intracellular SPM or SPD concentration (Figure 4a and Appendix A, respectively). Although SRM gene expression was consistently upregulated in SRS line 2 (Appendix A), quantitative Western blots revealed SRM protein level in this line was similar to that of the WT lines (Figure 4b). As AdoMetDC activity levels are essentially equal among the cell lines, the possibility exists for increased availability of dcAdoMet for SPD biosynthesis in the absence of SMS activity, thus contributing to SPD accumulation in these patients as well as the observed PUT depletion. In regard to SMS, a severe reduction (>90% less than the average WT *SMS* expression level) in the expression of the full-length *SMS* transcript was noted in SRS1 cells, consistent with some read-through of the mutated splice site previously reported in these cells (Appendix A) [11]. *SMS* transcript levels of the other two SRS lines were within the WT range.

### 3.4. Effects of Spermine Synthase Mutations on Polyamine Catabolism

One possible fate for excess SPD is its catabolism via SSAT (Figure 5a). A rate-limiting enzyme, SSAT catalyzes the transfer of an acetyl group from acetyl CoA to the *N*^1^ position of spermidine or spermine. These *N*^1^-acetylated polyamines are then either exported from the cell or oxidized by PAOX. This 2-step back-conversion via SSAT/PAOX thereby returns SPD or SPM to its precursor (PUT or SPD, respectively). As SPD and SPM induce SSAT expression at multiple levels [26,27], we investigated the possibility that SSAT mRNA and activity levels in the SRS lines might respond to the altered SPD/SPM ratios. Line SRS3 demonstrated elevated SSAT transcript levels as well as activity; however, a similar increase in activity was evident in WT line 2 (Figure 5b and Appendix A). Additionally, PAOX activity, which typically is dependent upon substrate availability, was significantly decreased by approximately 50% in each of the SRS lines compared to WT (Figure 5c). Overall, these data suggest that loss of SMS has little effect on the basal catabolism of SPD by SSAT, with the decreased PAOX activity in the SRS lines potentially serving to increase export of any SPD that does become *N*^1^-acetylated in lieu of its back-conversion to PUT. However, examination of culture medium removed from the lymphoblasts after 24 h of growth revealed a complete absence of polyamines, including the acetylated derivatives, suggesting that polyamine catabolism is not induced by excess SPD. SMOX, which directly oxidizes SPM back to SPD, was expressed at very low mRNA levels in nearly all of the lymphoblast lines (Appendix A), with no protein or activity detected regardless of *SMS* status.

### 3.5. Snyder-Robinson Syndrome Lymphoblasts Maintain Active Polyamine Transport

In addition to downregulating their own biosynthesis, an accumulation of polyamines also suppresses the polyamine transport system, thereby inhibiting the uptake of additional polyamines from the extracellular environment [21]. The lymphoblast cell lines were incubated in the presence of exogenous SPM to determine if polyamine uptake/transport was altered in SRS lymphoblasts. Treatment with 5 μM SPM for 24 h not only increased SPM levels in the SRS lines, but simultaneously decreased SPD levels (Figure 6a). With the exception of PUT, polyamine levels were effectively restored to those similar to WT SMS cells, indicating that the polyamine transport system was active and the SRS lymphoblast lines could self-regulate their polyamine pools in the presence of exogenous SPM. The SPD/SPM ratios of the SRS lines decreased from their baseline values (3.07, 7.35, and 9.31) to 1.11, 1.35, and 1.4, respectively (Figure 6b). SSAT activity following treatment with SPM was unchanged (Figure 6c), and HPLC analyses revealed a lack of *N*^1^-acetylated polyamines in both intracellular lysates and medium samples, indicating that SPD reduction in SRS cells is independent of SSAT induction.

To more accurately quantify the ability of the SRS lines to uptake polyamines, cells were incubated with the polyamine analog BENSpm, followed by HPLC analysis (Figure 6d). Overall, the WT and SRS lines were equally capable of accumulating BENSpm over 24 h. Line SRS1 accumulated the least BENSpm, but this was not significantly less than the WT line 1. Like biosynthesis, polyamine transport is negatively regulated by antizyme expression [21]; thus, the decreased uptake of BENSpm in SRS1 cells is likely associated with its increased antizyme level. Regardless of this decrease, the amount of SPM transported into each of the SRS cell lines was sufficient to reduce their intracellular SPD levels to within the range observed in the wildtype cells.

### 3.6. N^8^-Acetylation of Spermidine

Spermidine localized in the nucleus can undergo acetylation of its *N*^8^ position, which allows relocation of the spermidine moiety into the cytoplasm via charge neutralization. *N*^8^-acetylated spermidine was recently reported as a potential biomarker for SRS due to its elevated levels in the plasma of three SRS patients [28]. While the acetyltransferase responsible for this acetylation has yet to be definitively determined [29,30], it was recently found that deacetylation of *N*^8^-acetylspermidine to yield native spermidine is catalyzed by the cytoplasmic enzyme HDAC10 [31]. We therefore determined if HDAC10 expression levels were altered in the SRS lymphoblasts in our study. Although HDAC10 mRNA levels were significantly elevated in SRS line 2, it did not translate to an increase in protein observed via Western blot, and there was no overall difference between the wildtype and mutant SMS lines (Figure 7). Additionally, we failed to detect *N*^8^-acetylated spermidine in either the cell lysates or excreted into the culture medium, suggesting that lymphocytes were not the likely source of the plasma metabolite detected by Abela and colleagues [28].

### 3.7. Transglutaminase 2 Expression Is Upregulated in Snyder-Robinson Syndrome Patient Lymphoblasts

As low-molecular-weight amines, polyamines are natural acceptor substrates for the TG family of enzymes [8], which catalyze the calcium-dependent cross-linking of glutamine and lysine residues within or between proteins. TG activity also incorporates polyamines into certain glutamine residues of cellular proteins via one or both of their primary amino groups, thereby forming mono- or bis(γ-glutamyl)-PUT, SPD, or SPM, and potentially interfering with isopeptide bond cross-linking. To determine if the altered polyamine concentrations in the SRS lymphoblasts correspond to changes in TG, we analyzed the expression of transglutaminase 2 (TGM2), a ubiquitously expressed tissue TG (Figure 8). *TGM2* mRNA expression was significantly increased in all three SRS cell lines when compared to the WT cell lines, and TGM2 protein followed a similar trend, with the highest expression of both mRNA and protein in SRS line 2.

## 4. Discussion

We have investigated the differences in polyamine metabolism and transport in lymphoblastoid cell lines derived from three SRS patients and two wildtype donors. The SRS cell lines were chosen based on their extent of spermine deficiency, which ranged from approximately 50 to 25% of the WT lines. Both SPM and PUT levels were significantly decreased in the SRS lines relative to the WT lines, while SPD concentration was increased. The SPD/SPM ratio averaged 1.14 in the WT lines, while the SRS lines had elevated SPD/SPM ratios of approximately 3.76, 8.72, and 9.87, confirming previous observations [10,11,12,13]. 

The only significant change in gene expression between lymphoblasts derived from SRS patients versus controls was in *TGM2*. Although there was some variation in the levels of other genes investigated among the three SRS lines, the two WT cell lines also varied and there were no statistically significant differences between the groups. Analyses of enzyme activities and protein levels of polyamine-related genes did reveal changes between the SRS versus WT lymphoblasts. In particular, ODC activity was significantly decreased in all three SRS lines, which correlated with the decreased putrescine levels observed in these lines. Interestingly, an elevated level of ODC antizyme (OAZ1) was detected only in SRS line 1, which also displayed slightly decreased uptake of exogenous polyamines, consistent with the known ability of OAZ1 to downregulate the polyamine transport system. Thus, the decrease in ODC activity in the SRS cell lines is likely due, at least in part, to product inhibition of ODC by the high concentrations of SPD in the affected cells. The three SRS lines also had significantly reduced PAOX activity (~50%) compared to the WT lines, consistent with a lack of back-conversion to putrescine. 

The ability of the SRS lymphoblasts to normalize their intracellular polyamine levels upon treatment with exogenous spermine was impressive and highlights the potential of administering spermine as a therapeutic strategy. As ODC antizyme is stimulated by excess polyamines to negatively regulate polyamine biosynthesis and uptake, we anticipated that elevated antizyme in response to the elevated spermidine concentrations in the SRS patients that would prevent uptake via the polyamine transport system. This assumption was based on observations in the Gy mouse model, in which *Sms*-deficient male mice fed a diet containing SPM still had no detectable levels of SPM in brain, liver, or heart tissue [32]. However, studies using embryonic fibroblasts from these mice did indicate that spermine could be acquired from the culture medium [33]. In our study, although antizyme was induced in SRS line 1, its effect on polyamine transport was not sufficient to limit the maintenance of homeostasis in response to excess spermine. Although cell types more relevant to the SRS phenotype might differ in this regard, when considering systemic delivery of a therapeutic agent, the avoidance of off-target effects is also necessary. As the accumulation of excess polyamines is associated with certain pathologies, including cancer [4], the fact that these cells reestablish polyamine homeostasis rather than accumulating additional polyamines following SPM administration may have important consequences over the long term.

The finding that the lymphoblast cell lines derived from SRS patients exhibit levels of TGM2 that exceed those of normal donors may have significant clinical implications if conserved or exaggerated in cell types that contribute more to the SRS phenotype. Stimulation of peripheral blood lymphocytes with mitogen to induce blastogenesis induces the influx of Ca2+, which increases TGM activity as well as polyamine concentrations. However, even with mitogen stimulation, only a small number of lymphoblast proteins became conjugated with spermidine, and this binding was subsequently determined to be independent of transglutaminase activity [8,34]. Very few spermine-conjugated proteins were detected in mitogen-stimulated lymphocytes, and attempts to identify polyamines released from hydrolysates of acid-insoluble protein from our SRS lymphoblast lines have been unsuccessful. However, in other systems, such as seminal secretions, spermine becomes highly conjugated, and both spermine and spermidine are capable of cross-linking proteins via the incorporation of both primary amines [8]. As high affinity substrates for TG reactions, polyamines in excess, such as SPD in SRS patients, may serve as competitive substrate inhibitors of the cross-linking transamidation reactions, or they may themselves become incorporated into the cross-link when bound through both primary amines. Importantly, transglutaminases play roles in several processes associated with the characteristic phenotype of SRS males, including osteoblast, neuronal, and myoblast differentiation [35]. As it has been shown that transglutaminase activity responds to changes in ODC activity and polyamine concentrations, further investigation into its role in the SRS phenotype is warranted and ongoing [36]. 

Snyder-Robinson Syndrome studies using patient-derived material are limited by small patient number and acquisition of suitable cell lines for study. Lymphoblasts were chosen for this initial characterization, as they are most easily obtained from the patient and had been previously established. However, they do not correspond to a tissue that displays an obvious phenotype in SRS. These studies do, however, form a basis and provide knowledge for leading biochemical changes to study in affected tissues as well as for development of therapeutic strategies, particularly those aimed at replenishing intracellular spermine. Recently, a novel neurodevelopmental disorder was described resulting from gene variants that increase ODC activity [37,38]. Although, unlike SRS, these patients have elevated levels of ODC and putrescine, several clinical manifestations are similar, including hypotonia and developmental delays. Of note, with the exception of red blood cells, intracellular polyamine levels have not been reported in these patients. In a mouse model where ODC is specifically overexpressed in the skin, increased intracellular spermidine levels with concurrent decreases in spermine, relative to wildtype littermates were observed [39]. Thus, patients with *ODC1* gene variants may have intracellular SPD:SPM ratios similar to those of SRS patients. Similarly altered polyamine metabolism was also recently described in a mouse model of tuberous sclerosis complex (TSC), a neurodevelopmental disorder associated with mechanistic Target of Rapamycin Complex 1 (mTORC1) dysregulation. TSC also shares clinical manifestations with SRS, including intellectual disability and epileptic seizures, and inhibition of ODC reduced astrogliosis, an indicator of TSC neuropathology [40]. The common neurological phenotypes resulting from mutations in *SMS*, *ODC1*, and *TSC* and their association with dysregulated polyamine metabolism suggest that the knowledge obtained from the current studies might serve to benefit a patient population beyond those with Snyder-Robinson Syndrome. 

## Figures and Tables

**Figure 1 medsci-06-00112-f001:**
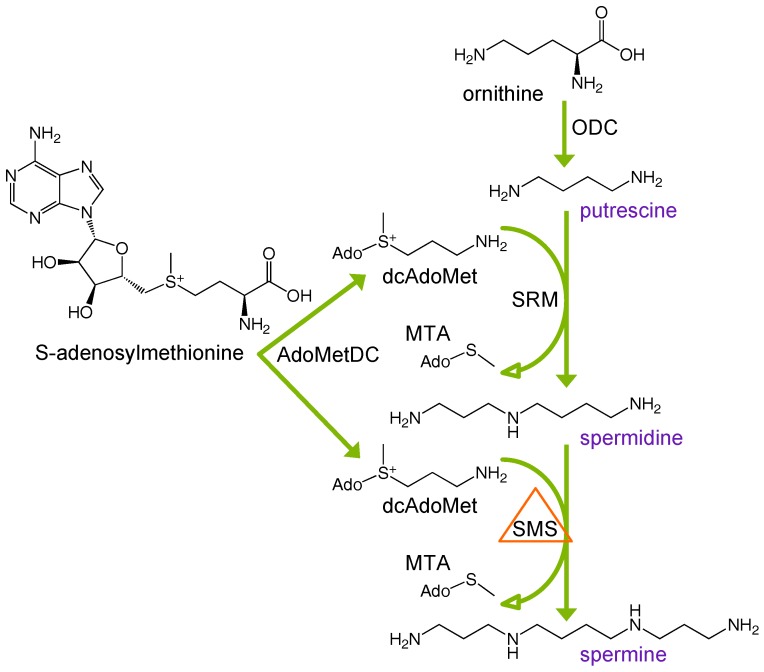
Mammalian polyamine biosynthesis. Polyamines are indicated in purple. Putrescine is created from ornithine via ornithine decarboxylase (ODC). Conversion of putrescine to spermidine and spermidine to spermine occurs through spermidine synthase (SRM) or spermine synthase (SMS), respectively. Both enzymes require the activity of S-adenosylmethionine decarboxylase (AdoMetDC) for the provision of the aminopropyl group donor (decarboxylated AdoMet, dcAdoMet). Snyder-Robinson Syndrome (SRS) patients are deficient in SMS activity, resulting in decreased spermine and accumulation of spermidine. MTA = methylthioadenosine.

**Figure 2 medsci-06-00112-f002:**
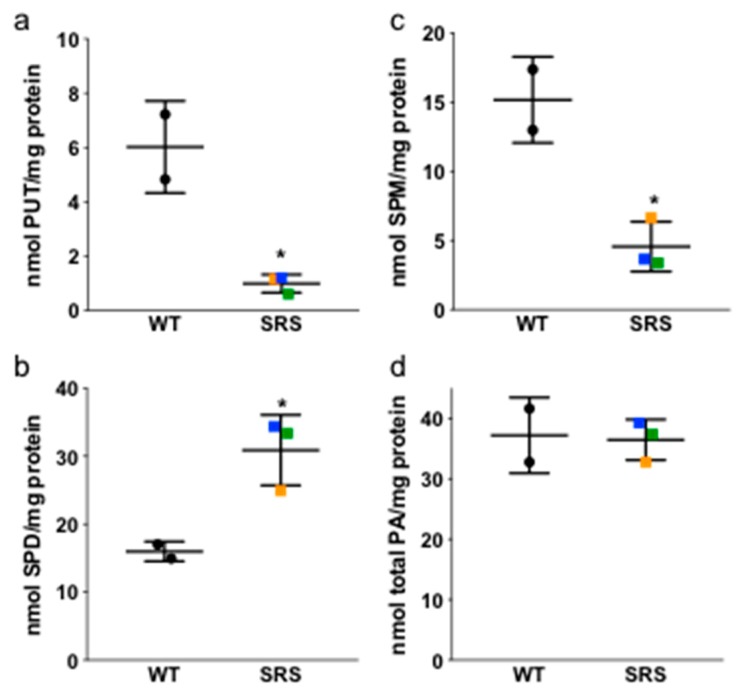
Alterations in basal intracellular concentrations of (**a**) putrescine (PUT), (**b**) spermidine (SPD), (**c**) spermine (SPM), and (**d**) total polyamines (PA) between *SMS* wildtype (WT) or mutant (SRS) lymphoblast cell lines (*n* = 5, each measured in duplicate). Concentrations are presented as nmol of polyamine per mg of cellular protein. The individual SRS cell line designations are orange for SRS1, blue for SRS2, and green for SRS3. Error bars indicate standard error of the mean (SEM). * *p* < 0.05.

**Figure 3 medsci-06-00112-f003:**
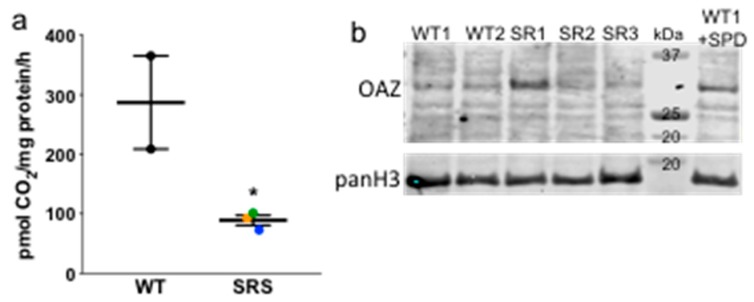
(**a**) ODC activity in donor or SRS lymphoblasts (*n* = 2, in triplicate; error bars = SEM), presented as pmol CO_2_ produced per hour per mg of total protein. Color designations are orange (SRS1), blue (SRS2), and green (SRS3). (**b**) Representative Western blot of ODC antizyme 1 (OAZ1) with pan histone H3 as loading control. The WT1 cell line treated with SPD for 24 h was used as a positive control for OAZ1. * *p* < 0.05.

**Figure 4 medsci-06-00112-f004:**
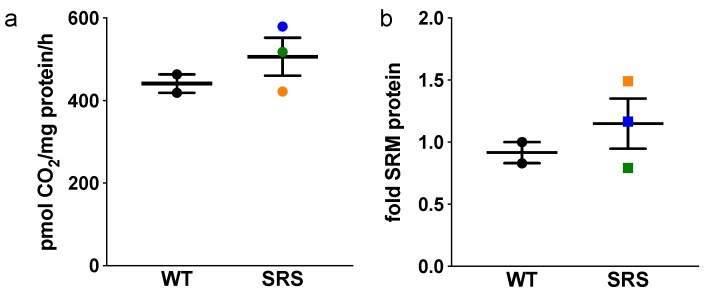
(**a**) Basal AdoMetDC activity (*n* = 2, in triplicate) of donor or SRS lymphoblast lines. Color designations are orange (SRS1), blue (SRS2), and green (SRS3). (**b**) Quantitative Western blots of SRM in lymphoblast cell lines (*n* = 2). All error bars indicate SEM.

**Figure 5 medsci-06-00112-f005:**
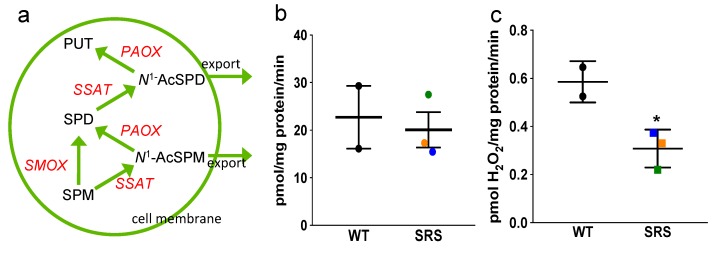
(**a**) The polyamine catabolic pathway. (**b**) Spermidine/spermine *N*^1^-acetyltransferase (SSAT) and (**c**) *N*^1^-acetylpolyamine oxidase (PAOX) activity assays. Color designations in (b,c) are orange for SRS1, blue for SRS2, and green for SRS3. Error bars indicate SEM (*n* ≥ 2, in triplicate; * *p* < 0.05).

**Figure 6 medsci-06-00112-f006:**
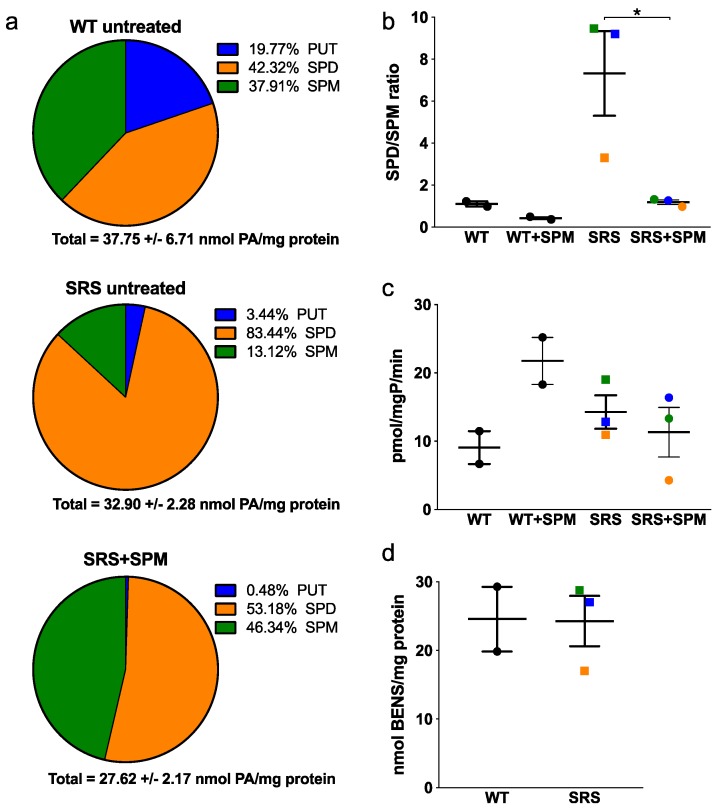
(**a**) Average polyamine levels of SRS lymphoblast lines before (middle) and after (bottom) 5 μM SPM treatment for 24 h, compared with untreated WT lymphoblast lines (top). (**b**) SPD/SPM ratios and (**c**) SSAT activity before and after SPM supplementation. (**d**) Intracellular accumulation of bis(ethyl)norspermine (BENSpm) following 10 μM treatment for 24 h. Color designations in (b–d) are orange for SRS1, blue for SRS2, and green for SRS3. All error bars indicate SEM (*n* ≥ 2, in triplicate; * *p* < 0.05).

**Figure 7 medsci-06-00112-f007:**
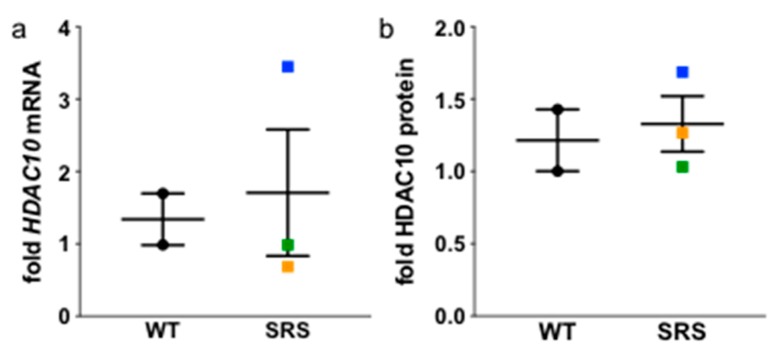
HDAC10 mRNA (**a**) and protein (**b**) levels in wildtype versus SRS lymphoblasts. Color designations are orange for SRS1, blue for SRS2, and green for SRS3. Error bars indicate SEM (*n* = 3).

**Figure 8 medsci-06-00112-f008:**
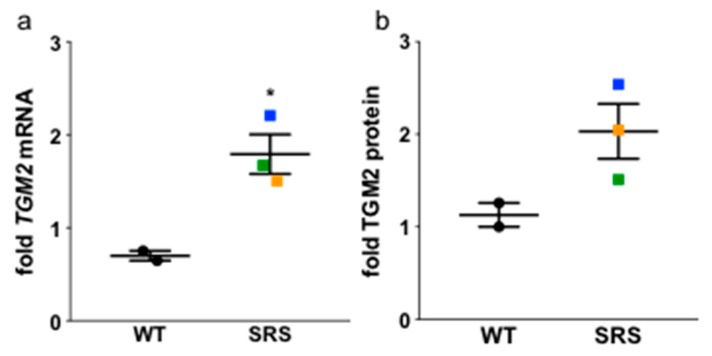
Transglutaminase 2 mRNA (**a**) and protein (**b**) expression levels are increased in SRS cell lines. Colors designate SRS line 1 (orange), SRS line 2 (blue), and SRS line 3 (green). All error bars indicate SEM (*n* = 3; * *p* < 0.05).

**Table 1 medsci-06-00112-t001:** Characteristics of lymphoblastoid cell lines. SPD/SPM ratios represent means with (SEM) *n* = 5. Mutations, protein products, and spermine synthase (SMS) activity were as previously reported [10]. ND = none detected.

Cell Line	Mutation	Protein	SMS Activity	SPD/SPM
WT1	none	wildtype	yes	1.17 (0.04)
WT2	none	wildtype	yes	0.83 (0.07)
SRS1	c.329+5 G>A aberrant splice site	truncated; some functional SMS from read-through	reduced	3.76 (0.25)
SRS2	V132G	decreased dimerization	ND	9.56 (0.73)
SRS3	G56S	no dimerization	ND	9.85 (0.36)

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
