# Peer review of "Polyamine Homeostasis in Snyder-Robinson Syndrome"

_medsci, 2018, doi:10.3390/medsci6040112_

Round 1

Reviewer 1 Report

The manuscript by Murray et al is an excellent contribution to the polyamine literature and specifically towards the better understanding of a rare condition referred to as Snyder-Robinson Syndrome (SRS). The authors are world experts in the field of study and have assembled a sound high quality manuscript with interesting preclinical type data that contribute toward attempts to design future therapies for SRS patients. The manuscript needs little to no revision and should be accepted in its current format.

Minor suggestions:

1) Consider specifying which antizyme (OAZ) the authors refer to. It is assumed to be OAZ1 but this should be stated clearly.

2) A recent publication in the American Journal of Medical Genetics by Bupp et al (2018) has described a second polyamine pathway-connected syndrome with a mutation in ODC1 that to increase ODC protein and putrescine. While this appears opposite of the herein described decrease of ODC and putrescine, some clinical manifestations are similar (intellectual disability, hypotonia) and this could be briefly discussed.

Author Response

Dear Reviewer 1,

Thank you for your positive feedback and suggestions to improve our manuscript. The following changes have been made:

Although we indicated OAZ1 in regard to the antibody and qRT-PCR primers in Materials and Methods, we have now changed all text references to "OAZ1" for clarity.

Your suggestion for adding a discussion of the similarities in phenotype between the SRS patients and those with ODC1 variants is greatly appreciated. It has now been added to the last paragraph of the Discussion section.

Reviewer 2 Report

The manuscript presents a comprehensive description of polyamine homeostasis in the SRS (Snyder-Robinson syndrome )-derived lymphoblasts which are deficient in spermine synthase. The authors have examined the effects of increased Spd/Spm ratios of SRS lymphoblasts on the polyamine biosynthetic enzymes (ODC, AdoMetDC), antizyme, catabolic enzymes (SSAT1 and PAOX) and polyamine transport activity at the levels of activity, mRNA and/or proteins and found a decrease in ODC and cellular putrescine in SRS-derived lymphoblasts. Interestingly, the levels of transglutaminase 2 (TG2) mRNA and protein were consistently increased in SRS-derived cells, suggesting a potential connection between polyamine homeostasis and TG2. The experiments were carefully conducted and the data support the conclusions. The manuscript is clearly written and should be acceptable for publication in Medical Sciences.

 Minor comments:

The authors may consider citing a related study (Wang, J-Y et al, 1994, Experimental Biology and Medicine, 205: 20-28)

A correction is recommended on description of the study of reference 8 (labeling of cellular proteins in lymphocytes incubated with radiolabeled putrescine, lines 310-313). That study inadvertently had misled the readers that the labeling of lymphocyte proteins was through transglutaminase reaction. However, it turned out not to be the case, but a new posttranslational modification leading to synthesis of hypusine in eIF5A (Proc. Natl. Acad. Sci. USA 78:2869-2873. PMID 6789324 PMC 319460). The three labeled proteins in Fig. 3 of Ref 8 are one 18K intact eIF5A and two degraded eIF5A fragments. The authors are correct in the statement on line 311 “ attempts to identify……..unsuccessful”.  In normal mammalian cells in culture, specific TGase substrate proteins could not be identified (probably too low to be detected by this method) by incubation in the presence of radioactive putrescine or spermidine. TGase modified proteins could only be detected by incubation of cells with biotinylated pentylamines and by subsequent western blotting.

Author Response

Dear Reviewer 2,

Thank you for your positive comments on our manuscript. According to your suggestions, we have made the following revisions:

We appreciate your clarification of the lymphoblast transglutaminase study by Folk et al. We have now edited this section to more clearly report the fact that few TG-mediated polyamine conjugates have been detected in lymphoblasts (lines 316-319).

The suggested reference has been added and discussed (lines 327-329), as it importantly demonstrates that TG activity responds to changes in polyamine levels. This may also have implications for our SRS patients, as some have reported very severe colitis. Thank you!